# Genetic Diversity and Population Structure of *Jubaea chilensis*, an Endemic and Monotype Gender from Chile, Based on SNP Markers

**DOI:** 10.3390/plants11151959

**Published:** 2022-07-28

**Authors:** Paola Jara-Arancio, Carolina da Silva Carvalho, Martín R. Carmona-Ortiz, Ramiro O. Bustamante, Priscilla M. Schmidt Villela, Sónia C. da Silva Andrade, Francisco T. Peña-Gómez, Luís A. González, Marina Fleury

**Affiliations:** 1Instituto de Ecología y Biodiversidad (IEB), Victoria 631, Barrio Universitario, Concepción 4030000, Chile; rbustama@uchile.cl (R.O.B.); lofhus@gmail.com (F.T.P.-G.); marina.fleury@mac.com (M.F.); 2Centro Internacional Cabo de Hornos (CHIC), Universidad de Magallanes, Teniente Muñoz 166, Puerto Williams 6350000, Chile; 3Departamento de Ciencias Biológicas y Departamento de Ecología y Biodiversidad, Facultad de Ciencias de la Vida, Universidad Andrés Bello, República 252, Santiago 8370134, Chile; 4Instituto Tecnológico Vale (ITV), Rua Boaventura da Silval, 955, Nazaré, Belém 66055-090, Brazil; carolina.carvalho@ymail.com; 5Facultad de Ciencias Biológicas, Pontificia Universidad Católica de Chile, Santiago 8331150, Chile; mcarmona@bio.puc.cl; 6Laboratorio de Ecología Geográfica, Departamento de Ciencias Ecológicas, Facultad de Ciencias, Universidad de Chile, Las Palmeras 3425, Ñuñoa, Santiago 7750000, Chile; 7EcoMol Consultoria e Projetos, Rua Ajudante Albano 847, Piracicaba 13416-030, Brazil; priscilla@ecomolconsultoria.com.br; 8Departamento de Genética e Biologia Evolutiva, IB-USP, Rua do Matão 277, São Paulo 05508-090, Brazil; soniacsandrade@ib.usp.br; 9Facultad de Ciencias Forestales y de la Conservación de la Naturaleza, Universidad de Chile, Av. Santa Rosa 11315, La Pintana, Santiago 8820000, Chile; algonzal48@yahoo.es; 10Projeto Origem, São Gonçalo do Amarante, São João del Rei 36300-970, Brazil

**Keywords:** genetic diversity, *Jubaea chilensis*, population structure, SNP, neotropical palm

## Abstract

*Jubaea chilensis* (Molina) Baill., also named Chilean palm, is an endemic species found in the coastal area of Mediterranean sclerophyllous forest in Chile. It has a highly restricted and fragmented distribution along the coast, being under intense exploitation and anthropogenic impact. Based on 1038 SNP markers, we evaluated the genetic diversity and population structure among six *J. chilensis* natural groups encompassing 96% of the species distribution. We observed low levels of genetic diversity, a deficit of heterozygotes (mean *H_E_* = 0.024; *H_O_* = 0.014), and high levels of inbreeding (mean *F_IS_* = 0.424). The fixation index (*F_ST_*) and Nei’s genetic distance pairwise comparisons indicated low to moderate structuring among populations. There was no evidence of isolation by distance (*r* = −0.214, *p* = 0.799). In the cluster analysis, we observed a closer relationship among Culimo, Cocalán, and Candelaria populations. Migration rates among populations were low, except for some populations with moderate values. The K value that best represented the spatial distribution of genetic diversity was ∆K = 3. Habitat fragmentation, deterioration of the sclerophyllous forest, lack of long-distance dispersers, and a natural regeneration deficit may have driven inbreeding and low levels of genetic diversity in the palm groves of *J. chilensis*. Although extant populations are not at imminent risk of extinction, the rate of inbreeding could increase and migration could decrease if the effects of climate change and human impact become more acute.

## 1. Introduction

Understanding the extent to which human activities affect the distribution of the genetic diversity of plant species is mandatory for geneticists and ecologists interested in the conservation of biodiversity [1]. High genetic diversity is necessary for species to cope with environmental changes and the ongoing genetic erosion, together with demographic aspects, can increase the extinction risk of natural populations [2]. Therefore, in the context of global change, the knowledge of genetic patterns in natural populations is essential for more effective conservation management of species. Habitat loss and fragmentation are considered human-induced disturbances of major importance for the genetics of natural plant populations [3]. The cascade effects of such disturbances may lead to reduced seed dispersal; increased isolation of populations; reduced species abundance; and, consequently, to bottlenecks and inbreeding in plant populations. The ensuing effects of genetic drift and reduced gene flow may place some species at the edges of extinction [4]. Animal-dispersed plants can be particularly affected by habitat loss and fragmentation, because large seed disperser vertebrates are particularly prone to extinction [5]. As these vertebrates can move seeds over long distances [6], the extinction of frugivores may lead to an accumulation and concentration of seedlings of plants that depend on animals for seed dispersal near the parent plants. The reduction of long-distance seed dispersal may augment genetic structure within and between seedlings of plants that depend on anween populations and decrease the genetic diversity at the population level [7,8,9,10].

Palms (Arecaceae) were among the first plant groups to receive attention regarding the risks of becoming endangered [11], but despite this early attention, many palm species are heading toward population collapse (26). Today, the IUCN Red List of Threatened Species includes about 31% of the world’s 1150 palm species. The interest in palm conservation stems from both their economic [12,13] and ecological importance [14,15]. Palms have been regarded as ‘keystone resource species’ [14,15], as their disappearance may have a cascade effect on natural ecosystems, changing the competitive relationships and relative abundance of other species in the community [16,17]. From a socioeconomic perspective, palms are considered a prime non-timber forest product (NTFP) resource [12], as its fruits, leaves, stems, seeds, sap, and other parts are exploited, destructively or not, for numerous purposes [18,19,20].

The coastal sclerophyllous forest, located at the Chilean Mediterranean biome in Central Chile (31°52′–37°20′ S), is a diverse ecosystem with high levels of endemism [21,22,23]. The *palmar*, or palm forest, is an overlooked component of the sclerophyllous forest, which occurs in patches with individuals of the long-lived *Jubaea chilensis* (Molina) Baill., the Chilean palm, a tertiary-relict endemic species [24]. The rainforest contractions in Central Chile during the late Pleistocene and the extensive use of Chilean palm, combined with intense human activity for land change uses in Central Chile, have reduced the pre-Columbian populations of *J. chilensis* to 2.5% [25,26]. More recently, such changes are mostly driven by habitat loss and fragmentation, xylem sap extraction from decapitated palm trees, illegal seed overharvesting for human consumption, and recurrent wildfires. All of these synergic disturbances are thus expected to have detrimental consequences on the *J. chilensis* genetic diversity [3]. *Jubaea chilensis* is an arborescent, woody monocot species and is diclino-monoecious. The fruit is a drupe with an orange fleshy pulped-pericarp and a hard-lignified endocarp [27,28,29,30]. This palm species presents a low frequency of long-distance seed dispersal events, reduced survival of seedlings, and its habitat has high anthropogenic intervention [25,26] In the Pleistocene, seed dispersal was the product of endozoocoria by the extinct megafauna species belonging to the following families: Gomphotheriidae, Camelidae, Equidae, Notohippidae, Homalodotheriidae, Toxodontidae, Astrapotheriidae, Macraucheniidae, Mylodontidae, and Megatheriidae [26,31]. Currently, the seeds are mainly dispersed in territories close to the mother plant as a result of gravity, floods, and runoff [26,32,33]. Humans can transfer the seeds to different commercialization sites, and the *Octodont degus* Molina and *Spalacopus cyanus* Molina rodents hoard the seeds for later consumption [26]. The frequency of rodent–seed interaction is low (<25%) and the transport usually occurs over short distances (<6 m, or 6.2 m [34,35]), indicating that rodent-dependent dispersion may be insufficient for genetic exchange between existing palm groves, which leads to a lack of gene flow and isolation between patches [26]. This precariousness in dispersion is highly affected by the black rat *Rattus rattus* L. that predates the seeds. Cordero et al. [34] found that the germination rate was 6%, plant survival was 1.81%, and only 7.9% of surviving seedlings become infantile plants (4 years old). The recently introduced domestic species, such as cattle and horses, also have an important effect, considering that they remove, consume, and regurgitate the seeds, but leave them susceptible to competition and desiccation [26,35]. In spite of the fact that pollen dispersal can cover a greater distance than seeds [36,37,38], in this case, its movement is limited because of the abundant flowers produced by just one individual, which mean that pollinators visit contiguous flowers of the same individual, and visits to the flowers of other individuals are very infrequent events [26]. It has been described that, in systems with restricted pollen dispersal, the interruption of genetic connectivity can occur even in continuous populations [39,40,41].

To assess the magnitude of the loss of genetic diversity that *J. chilensis* populations present as a result of isolation and low population sizes, a genetic conservation study was carried out in remnant populations of the species under study. Specifically, we estimated parameters of genetic diversity and inbreeding and evaluated genetic structure and patterns of contemporary migration among populations. The information obtained will be crucial to understand the current genetic diversity as well as provide input for planning strategies for safeguarding sustainable management of the conservation of this emblematic species.

## 2. Results

### 2.1. Statistical Analysis and Genetic Diversity

The total number of reads was 226,279,541 with good average quality (Q-score ≥ 30), showing an optimal value (Q ≥ 30) throughout all of the sequences (101 bp, –read length sequenced by the Illumina HiSeq platform). A total of 1038 SNPs were selected for this study, according to our previously established parameters. The distribution of SNP minor allele frequencies (MAFs) is shown in Appendix A. More than 96.92% of the SNPs for all sampled populations presented MAF values ≤ 0.1, varying from 0.994 in Culimo to 0.969 in Viña del Mar/Valparaíso, with MAF scores ranging up to MAF ≤ 0.4.

Expected heterozygosity values (*H_E_*) were higher than the observed values (*H_O_*) for all populations. The mean *H_O_* varied from 0.036 in Culimo to 0.108 in Petorca, and the mean *H_E_* varied from 0.086 in Culimo to 0.145 in Petorca, with an overall *H_O_* of 0.014 and an overall *H_E_* of 0.024, both suggesting a deficit of heterozygotes. Inbreeding coefficient values (*F_IS_*) for all populations were statistically significant (overall value of 0.424, with a 95% confidence interval not including zero), varying from 0.177 in Viña del Mar/Valparaíso to 0.586 in Culimo, which also indicates high levels of inbreeding with little or no random mating (Table 1). All *F_ST_* comparisons between populations showed values above 0.049 and below 0.119, which indicates moderate genetic structuring (Table 2). Nei’s genetic distance values were lower than 0.003 in all populations, also indicating that the populations may be genetically similar (Table 2). In the PCO, the first two axes explained 61.4% of the variance (35.0% and 26.4%, respectively). There was relative proximity between the Culimo, Cocalán, and Candelaria group and the Petorca population, while the Ocoa and Viña del Mar/Valparaíso populations remained in opposite peripheries (Figure 1). Mantel test results indicated that Nei’s genetic distance matrix did not significantly correlate with the geographic distance matrix (*r* = −0.214, *p* = 0.799), i.e., the nearest populations did not have lower values of genetic differentiation. When observing the Mantel analyses between geographical distance and *F_ST_*, similar results were observed, that is, there was no significant correlation (*r* = −0.125, *p* = 0.669). Low migration rates were observed (m = 0.010–0.022), except for the following populations (source receiver): Cocalán Candelaria (0.237 [0.181–0.293]), Cocalán Culimo (0.280 [0.238–0.322]), Cocalán Petorca (0.266 [0.215–0.317]), and Viña del mar-Valparaiso Ocoa (0.268 [0.222–0.314]) (Table 3). The log likelihood values were comparable between BA3-SNPs runs, as was the Bayesian deviance (Appendix A).

### 2.2. Genetic Structure and Admixture Levels

Based on the maximum value of ΔK, the estimated number of optimal groups was K = 3 (Figure 2 and Appendix A). Individuals from the Cocalán, Culimo, Candelaria, and Petorca populations had very close genetic characteristics, while Viña del Mar/Valparaíso was characterized as a unique group for almost all K values. The Culimo and Petorca individuals remain together for all of the assigned K values (Figure 2). At K = 3, 99% of the individuals in each population are correctly assigned to the area from which they were sampled, and only one individual from OCOA (129) contains attributions higher than 50% from other areas of origin (Appendix A).

## 3. Discussion

Our results show that all studied populations of *J. chilensis* present low genetic diversity and high inbreeding. Given their biallelic heritage, the highest value of H_E_ expected for SNPs is 0.5. However, the highest value observed for *J. chilensis* was less than a third of that value, indicating that these populations likely suffer from loss of genetic diversity. Null alleles are an issue for many marker types and could also result in a downward bias in estimated heterozygosity. Unfortunately, we are not able to estimate null alleles’ frequency and SNP null alleles have not been described in this species. The inbreeding coefficient was significant and high for all populations, even though the species is diclino-monoescious, which favored crossing [42]. Interestingly, the population with a small number of specimens (CUL) had the lowest value of H_E_ and the highest value of *F_IS_*. All of these results are indicative that *J. chilensis* may be suffering from genetic erosion. Other palm species have also shown high levels of inbreeding, and it was usually associated with the mating system, anthropic effects, and low abundance of populations [43,44,45,46,47,48]. The levels of inbreeding found in this study are concordant with what has been found in other species of the Arecaceae family. For instance, Shapcott et al. [43] found high levels of inbreeding in populations of five species of the genus *Pinanga*. Cibrián-Jaramillo et al. [44] found high inbreeding in populations of *Chamaedorea ernesti-augusti*, and declare that the high inbreeding is a result of the anthropic effect. Santos et al. [45], using nine ISSR markers, showed high levels of inbreeding and low abundance levels of *Attalea vitrivir* individuals. Other studies using microsatellites confirmed the prevalence of inbreeding in *Acrocomia aculeata* populations, product of crosses between very close relatives, or even full siblings [46,47,48]. For *Syagrus coronata*, for example, the authors take into account the high-level of inbreeding to conduct conservation measures [49]. Given that the sclerophyllous forest inhabited by *J. chilensis* has been highly exploited and approximately only 121,284 specimens are found in Central Chile (2.5% of the original population of the 19th century), being distributed among highly fragmented populations [26,50,51], conservation measures for this species should consider the low level of genetic diversity and inbreeding found in our study.

Contrary to our expectation, we found low genetic structure, distant populations of *J. chilensis* being genetically similar, and no evidence of isolation by distance. These results are unlikely to reflect the current seed dispersal pattern of the species. Currently, seeds of *J. chilensis* are dispersed close to mother plants, as rodents (*Octodont degus* and *Spalacopus cyanus*) and introduced domestic species are the main seed dispersers. The frequency of rodent–seed interaction is low (<25%) and transport usually occurs over short distances (<6 m [35]), indicating that rodent-dependent dispersion may be insufficient for genetic exchange between existing palm groves [26]. Pollen dispersion is also unlikely to explain the low genetic structure among the populations found here, as the movement of pollen between individuals of *J. chilensis* seems to be limited owing to the high abundance of flowers produced per individual [26]. It has been described that, in systems with restricted pollen dispersal, the interruption of genetic connectivity can occur even in continuous populations [39,40,41].

Nowadays, the geographical distribution of palm groves is highly scattered in a series of small, isolated patches, consisting of small independent local populations at a geographic level, as depicted in Figure 1. As a consequence, the degree of inbreeding increased as a result of the crossing between relatives [52,53,54]. Studies on different taxa of the Arecaceae family suggest that fragmentation considerably affects the persistence of palm trees. In the south of the Brazilian Amazon, forest fragmentation has affected *Bactris* Jacq. ex Scop. genus (palm), leading to inbreeding depression and possibly leading to extinction of the local palm tree [55]. For the *Astrocaryum aculetassimum* (Schott) Burret, an endemic Atlantic Forest palm species, it has been reported that the species populations are highly affected by the loss of seed dispersers as a result of fragmentation and hunting [56]. Moreover, for the *Astrocaryum mexicanum* Liebm. ex Mart. species inhabiting Los Tuxtlas (State of Veracruz/México), the abundance of coleopterans and pollinating beetles varies according to the size of the fragment, making reproduction doubly susceptible to the effects of fragmentation [57]. In Ecuador, the seedlings of *Ceroxylon echinulatum* Galeano and *Attalea colenda* (O.F. Cook) Balslev and A.J. Hend. have failed to survive as a result of deforestation [37,58,59]. An extensive review of the literature of palms from Tropical America indicated that anthropic influences may cause changes in the genetic structure, increasing inbreeding, and genetic drift in fragmented populations [37]. Similarly to this study, the *Phoenix dactylifera* L. populations occurring in natural oases in Tunisia also presented H_E_ values higher than H_O_ values with low genetic structure [60].

It is important to highlight that genetic structure analysis may reflect both contemporary and historical processes such as past gene flow and population fluctuation [61], and that the contemporary migration rates estimated here usually correspond to the last three generations. Moreover, *J. chilensis* is a palm tree with a significantly long life expectancy, with 1000 years being the estimated age for the oldest individual [29,62]. Therefore, the low genetic structure detected for some populations and the moderate migration rates observed among some populations could be a result of high gene flow in the past. In the Pleistocene, seeds of *J. chilensis* were consumed and dispersed by the extinct megafauna species belonging to the following families: Gomphotheriidae, Camelidae, Equidae, Notohippidae, Homalodotheriidae, Toxodontidae, Astrapotheriidae, Macraucheniidae, Mylodontidae, and Megatheriidae [26,31]. Simulations suggest that extinct megafauna would frequently disperse large seeds over a long distance, and the events of long-distance seed dispersal by these animals would be up to ten times longer than long-distance dispersal by smaller-sized extant mammals [63]. Cocalán is the second largest population; this can be related to the fact that Cocalán presents an excellent habitat for the growth and development of Jubaea, where the soils are almost exclusively granitic [29] and highly resistant to drought and shade [64].

Germination and establishment studies of *J. chilensis* have been carried out [65,66,67,68] in the context of the National Plan for the Conservation of the Chilean Palm (Corporación Nacional Forestal, CONAF) in 2005. Unfortunately, the Plan does not have recommendations to evaluate the genetic structure of this palm as a fundamental requirement to propose viable and efficient actions for the conservation of this species [1]. As a result, active restoration of *J. chilensis* via sowing seedlings has been carried out without any consideration of genetic or biogeographic origin. Understanding the distribution of genetic variability can be especially important for species facing extinction risk, especially when translocations for restoration, genetic rescue, or assisted gene flow are fundamental aspects for the successful long-term population [69]. Currently, the Chilean palm is listed as endangered (EN) by IUCN. The lack of long-distance seed dispersal events, no capacity for vegetative propagation, and low seed regeneration, plus its severe seed harvest and demographic bottleneck, may increase the extinction risk of *J. chilensis* capacity [26,60,61]. Although at this moment there is no imminent risk, all these factors could promote genetic isolation, either within or between populations, which can intensify the loss of genetic diversity and inbreeding depression [41,54].

## 4. Conclusions

All populations of *J. chilensis* studied have low genetic diversity, high inbreeding, and no evidence of isolation by distance. Habitat fragmentation, deterioration of the sclerophyllous forest, lack of long-distance dispersers, and natural regeneration deficit may have driven inbreeding and low levels of genetic diversity in the palm groves of *J. chilensis*. Although extant populations are not at imminent risk of extinction, the rate of inbreeding could increase, and the migration effect might be overwhelmed by the effects of climate change and human impact. Thus, considering our findings plus the natural history and ecological context wherein *J. chilensis* lives, we suggest that this monospecific, relict, and endemic palm species should be considered as critically endangered by international conservation organizations.

## 5. Material and Methods

### 5.1. Study Organism

*Jubaea chilensis* is an arborescent and woody monocot, with a bare and cylindrical stipe narrower towards the top, reaching up to 30 m in height and from 0.80 to 1.10 m in diameter. It is a diclino-monoecious plant with cross pollination [42] and the fruit is a drupe with a single spherical seed of approximately 2–3 cm (0.79–1.18 pol.) in diameter. The fruit has an orange fleshy pulped-pericarp and a hard-lignified endocarp [26,27,28,29,30]. *J. chilensis*’ natural populations are distributed from La Serena (29°54′ S–71°15′ W) in the Coquimbo Region to Tapihue-Pencague (35°15′ S–71°47′ W) in the Maule Region. It inhabits warm climate areas with dry summers and coastal areas with coastal fog influence [29]. The species is found from sea level to approximately 1400 m [58], thus tolerating temperatures from 2.9 °C to 30.8 °C, with precipitations ranging from 127 to 879 mm [26,63]. The remaining populations are in an advanced stage of aging, with little or no appearance of natural regeneration and under a high anthropic impact due to the commercialization of their fruits and sap [26]. The dispersed seeds germinate for up to four years, forming persistent seed banks. Although the species adapts very well to its environment, the first stages of growth in which the seedlings must survive under the canopy are critical [62], especially in the understory of sclerophyllous and/or spiny species [63,64].

### 5.2. Study Area and Sampling

For the selection of population groups and individuals of *J. chilensis* to be sampled, we considered four parameters: (*i*) populations with more than 50 individuals; (*ii*) a sampling scheme that covers the entire species’ geographical range; (*iii*) the exclusion of specimens belonging to plantations; and (*iv*) a maximum of 30 individuals per population was analysed. We sampled the following populations groups: (a) Culimo (CUL), (b) Petorca (PET), (c) Ocoa (OCO), (d) Viña del Mar/Valparaíso (VAV), (e) Cocalán (COC), and (f) Candelaria (CAN) (Figure 1, Table 1). These groups of populations correspond to 95.98% of the total species abundance and are differentiated mainly by the orographic and edaphoclimatic characteristics (Figure 3, Table 4).

### 5.3. Genotyping-by-Sequencing Library and Single Nucleotide Polymorphisms’ Selection

We collected leaf tissue samples from 140 adult individuals of *J. chilensis* chosen at random, which included the entire distribution range and with a minimum separation between individuals of 150 m, between February and December 2015 (Table 4). Samples were stored in silica. For molecular analysis, DNA was extracted using the DNeasy Plant Mini Kit (Qiagen, EUA). DNA concentration was quantified using the Qubit High Sensitivity Assay kit (Invitrogen) and DNA integrity was assessed through visualization in a 1.2% agarose gel electrophoresis. Prior to library construction, DNA amount per sample was normalized at 100 ng/μL.

The Genotyping-by-Sequencing (GBS) library was constructed using the standard protocol described by Elshire et al. (2011, [70]) and employing the *ApeK*I restriction enzyme. Single-end 100-bp sequencing was conducted on the Illumina HiSeq 2500 platform. Samples were demultiplexed according to their respective barcodes. The generated sequences were filtered to remove low-quality sequences and contaminated reads using SeqyClean 1.9.9 (https://github.com/ibest/seqyclean, accessed on 1 January 2021); only high-quality paired-end sequences (with average PhredScore over 24 and over 65 bp) were used for further analysis. The single nucleotide polymorphisms’ (SNPs) prospection was performed using the software pipeline PyRAD v1.2 [71]. Reads with more than five Ns or shorter than 35 bp were discarded. The clustering threshold was set to 90% and the maximum number of SNPs per locus was set to 30. A locus had to be present in at least 50% of the samples to be retained in the final dataset. All other parameters were maintained at default values. To conduct population genetic analysis, we identified putative loci under selection using software BayeScan 2.1 [72] using the default values (Q-value < 0.05). The LOSITAN software was also used to identify loci under selection [73] and later removed from the SNPs dataset. The R package r2vcftools (https://github.com/nspope/r2vcftools, accessed on 1 January 2021)—a wrapper for VCFtools [74]—was used to perform final quality control on the genotype data. Filtering criteria included biallelic SNPs, linkage disequilibrium (r^2^ < 0.8) [75], Hardy–Weinberg equilibrium (HWE, *p* > 0.0001), and loci with less than 20% missing data.

### 5.4. Genetic Analysis

The fixation index coefficient (*F_IS_*) was calculated based on the variance found in the allelic frequencies [76], and the intra-population genetic diversity (expected and observed heterozygosities) was estimated under the Hardy–Weinberg equilibrium. Both analyses were performed using the ‘het’ option in VCFtools implemented in r2vcftools (https://github.com/rojaff/LanGen_pipeline, [74]—accessed on 1 January 2021). *F_ST_* calculations between populations were performed using the dartR package v.183 [77] in R software (R Core, [78]). Genetic distance was estimated according to Nei (1972), where a constant and equal mutation rate is considered for all loci with equal population sizes for all generations and a mutation–drift balance, using the adegenet package in R [79]. Additionally, the geographical distance (in meters) between pairs of populations was calculated considering the Earth’s curvature (assuming the spherical model), using the geosphere package in R (R Core, [80]). Mantel test based on Spearman’s correlation with 9999 permutations was performed using geographical distance with *F_ST_*, using the vegan package [81]. A principal coordinate analysis (PCO) based on Nei genetic distance was conducted using the pcoa command of the ape package [82,83]. We also calculated the migration rates (m) [84] of the studied populations. We used the BA3-SNPs v 3.0.4 [85] to determine the amount and direction of migration between populations. BA3-SNPs was run ten times using a different random seed across 10,000,000 iterations with a burn-in of 1,000,000 iterations and sampling every 1000 iterations. To compute the suitable allele frequency (a), inbreeding coefficient (f), and migration rate (m), we tested several values until we obtained the acceptance percentages recommended by BA3-SNPs authors (between 20 and 60%), being a—0.80, m—0.60, and f—0.60. We then assessed model convergence across all ten runs using Tracer v1.7.2 (http://beast.community/tracer, accessed on 1 January 2022). We used the Bayesian deviance as calculated by Meirmans (2014, [86]) to search for the best fitting model (the one with the lowest Bayesian deviance was selected for interpretation) [87]. A rough 95% confidence interval (CI) was constructed as mean ± 1.96 * sdev. All migration rates whose 95% confidence intervals did not include zero were reported as significant.

### 5.5. Genetic Structure and Admixture Levels

Populational structure and admixture degree were inferred using the Bayesian grouping method implemented in the STRUCTURE v 2.3.4 program [88,89,90]. Parameters were calculated for each K value (1 ≤ K ≤ 6) by means of an MCMC analysis of 100,000 replicates with a burn-in period of 20,000. The results were obtained based on 25 runs per population grouping, assuming both the Admixture model, which allows individuals to have multiple population origins, and the correlated allelic frequency model [88,89]. Finally, the posterior probability for each K value was determined by introducing the simulation results into STRUCTURE HARVESTER v 0.6.94 [91], where K values were evaluated using the logarithm of likelihood of the observed data (LnP [D]) as well as the second-order change rate of the data s log-likelihood in distinct runs of K (ΔK) [92].

## Figures and Tables

**Figure 1 plants-11-01959-f001:**
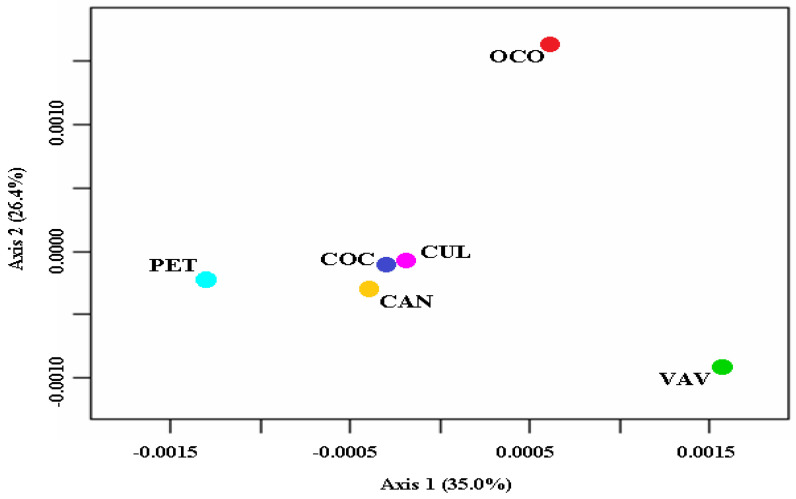
Plot of the principal coordinate analysis based on Nei’s genetic distances for the six sampled *Jubaea chilensis* populations, using the pcoa command in the ape package (Paradis and Schliep, 2019). (CUL) Culimo, (PET) Petorca, (OCO) Ocoa, (VAV) Viña del Mar/Valparaíso, (COC) Cocalán, (CAN) Candelaria.

**Figure 2 plants-11-01959-f002:**
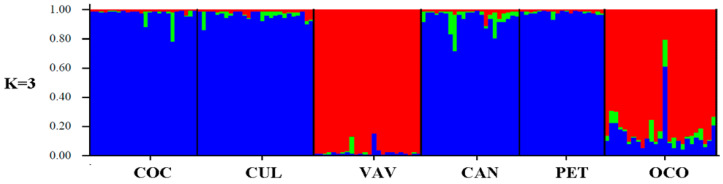
Inferred population structure for the *Jubaea chilensis* six population groups in central Chile based on an analysis of 1038 SNPs using STRUCTURE v 2.3.4, under the Admixture model. Each individual is represented by a vertical bar, often partitioned into coloured segments, with the length of each segment representing the proportion of the individual’s genome from K ancestral populations. (CUL) Culimo, (PET) Petorca, (OCO) Ocoa, (VAV) Viña del Mar/Valparaíso, (COC) Cocalán, (CAN) Candelaria.

**Figure 3 plants-11-01959-f003:**
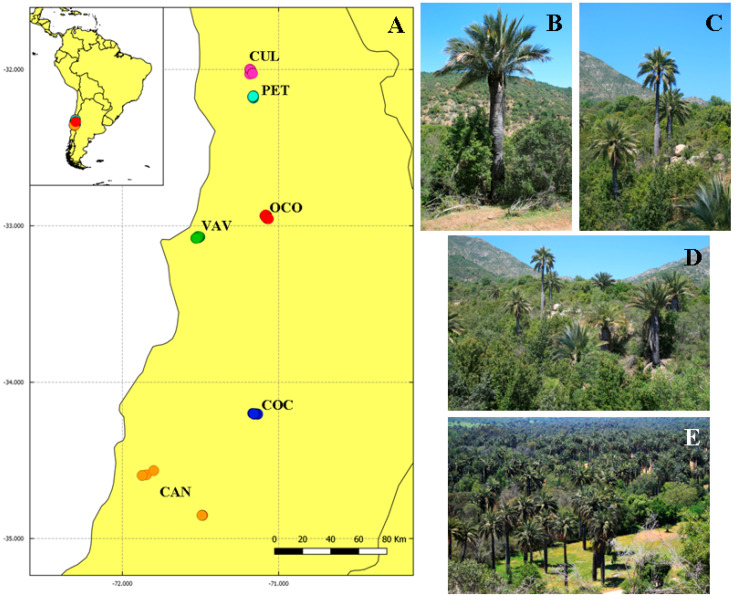
Characterization of *Jubaea chilensis* populations and sampling sites. (**A**) Location of the six sampled population groups in central Chile. (**B**,**C**) *Jubaea chilensis* habitat. (**D**,**E**) Hillside and valleys habitats where palm groves are distributed. The acronyms of the population groups are as follows: Culimo (**CUL**), Petorca (**PET**), Ocoa (**OCO**), Viña del Mar/Valparaíso (**VAV**), Cocalán (**COC**), and Candelaria (**CAN**).

**Table 1 plants-11-01959-t001:** Genetic diversity of the six population groups of *Jubaea chilensis* in Central Chile. (Sample no) number of sampled individuals, (*H_O_*) mean observed heterozygosity, (*H_E_*) mean expected heterozygosity, (*F_IS_*) mean inbreeding coefficient; (CUL) Culimo, (PET) Petorca, (OCO) Ocoa, (VAV) Viña del Mar/Valparaíso, (COC) Cocalán, (CAN) Candelaria. Values inside square brackets represent the 95% confidence interval.

Populational Grouping	Sample No	*H_O_* [95% CI]	*H_E_* [95% CI]	*F_IS_* [95% CI]
**CUL**	26	0.036 [0.027–0.045]	0.086 [0.086–0.087]	0.586 [0.484–0.689]
**PET**	19	0.108 [0.090–0.127]	0.145 [0.144–0.145]	0.251 [0.123–0.379]
**OCO**	25	0.065 [0.058–0.073]	0.101 [0.100–0.101]	0.351 [0.276–0.425]
**VAV**	24	0.107 [0.098–0.116]	0.130 [0.130–0.131]	0.177 [0.109–0.244]
**COC**	24	0.061 [0.051–0.070]	0.105 [0.104–0.106]	0.420 [0.330–0.510]
**CAN**	22	0.045 [0.039–0.051]	0.107 [0.106–0.107]	0.581 [0.528–0.633]
**Overall**	**140**	**0.014 [0.012–0.015]**	**0.024 [0.023–0.024]**	**0.424 [0.374–0.475]**

**Table 2 plants-11-01959-t002:** Weir and Cockerham fixation index (*F_ST_*) and genetic distance (Nei 1972) pairwise among six population groups of *Jubaea chilensis* in Central Chile. *F_ST_* values are below the diagonal and genetic distance of Nei (1972) values are over the diagonal. (CUL) Culimo, (PET) Petorca, (OCO) Ocoa, (VAV) Viña del Mar/Valparaíso, (COC) Cocalán, (CAN) Candelaria. * *p* < 0.05.

	CUL	PET	OCO	VAV	COC	CAN
**CUL**		0.0018	0.0020	0.0021	0.0015	0.0017
**PET**	0.078 *		0.0028	0.0030	0.0023	0.0025
**OCO**	0.054 *	0.080 *		0.0027	0.0024	0.0026
**VAV**	0.072 *	0.113 *	0.065 *		0.0026	0.0027
**COC**	0.063 *	0.119 *	0.074 *	0.100 *		0.0019
**CAN**	0.049 *	0.078 *	0.059 *	0.078 *	0.068 *	

**Table 3 plants-11-01959-t003:** Mean migration rates estimated by BayesAss 3.0.4 with 95% confidence intervals in square brackets. The values above are estimated values and the values below are the confidence interval. (CUL) Culimo, (PET) Petorca, (OCO) Ocoa, (VAV) Viña del Mar/Valparaíso, (COC) Cocalán, (CAN) Candelaria.

Receiver/Source	CUL	PET	OCO	VAV	COC	CAN
**CUL**	**0.677** **[0.657–0.696]**	0.010 [−0.009;0.030]	0.010 [−0.009;0.031]	0.010 [−0.009;0.030]	**0.280** **[0.238;0.322]**	0.010 [−0.009;0.030]
**PET**	0.013 [−0.011;0.037]	**0.680** **[0.654–0.705]**	0.013 [−0.012;0.038]	0.013 [−0.011;0.038]	**0.266** **[0.215;0.317]**	0.013 [−0.011;0.038]
**OCO**	0.010 [−0.009;0.031]	0.010 [−0.009;0.030]	**0.688** **[0.659–0.716]**	**0.268** **[0.222;0.314]**	0.010 [−0.009;0.029]	0.010 [−0.009;0.030]
**VAV**	0.011 [−0.009;0.031]	0.011 [−0.009;0.032]	0.011 [−0.009;0.031]	**0.932** **[0.884–0.981]**	0.022 [−0.008;0.053]	0.011 [−0.010;0.032]
**COC**	0.011 [−0.010;0.032]	0.011 [−0.009;0.032]	0.011 [−0.010;0.032]	0.010 [−0.009;0.031]	**0.944** **[0.900–0.987]**	0.011 [−0.010;0.032]
**CAN**	0.011 [−0.010;0.034]	0.012 [−0.010;0.034]	0.011 [−0.010;0.034]	0.011 [−0.010;0.034]	**0.237** **[0.181;0.293]**	**0.714** **[0.672–0.757]**

**Table 4 plants-11-01959-t004:** Geographical characterization of the six populations of *Jubaea chilensis* sampled. (XO) Xeric Oceanic Mediterranean climate. (PO) Oceanic Pluviseasonal climate. (Lat/Long) Latitude/longitude. The registered coordinates correspond to the average of the sampled population groups (*sensu* [26]).

Population Groups	Code	Lat/Long Coordinates	Climate	No. Specimens	No. Sampled Individuals
**Culimo**. Monte Aranda, Culimo, and El Naranjo	CUL	32°00′–71°11′	XO	204	26
**Petorca**. Túnel de Las Palmas and Las Palmas de Pedégua	PET	32°09′–71°09′	XO	1300	19
**Ocoa**. Parque La Campana, Hacienda Las Palmas del Ocoa, Oasis La Campana and Vichiculén-Llay Llay	OCO	32°57′–71°04′	PO	70,308	25
**Viña del Mar/Valparaíso**. Palmar Hacienda las Siete Hermanas and Subida Santos Ossas	VAV	33°04′–71°31′	PO	7200	24
**Cocalán**. Hacienda Las Palmas de Cocalán and La Palmería	COC	34°12′–71°08′	PO	35,500	24
**Candelaria**. Palmar de Candelaria	CAN	34°51′–71°29′	PO	1900	22

## Data Availability

The SNPs will be deposited in Figshare and a url address will be provided upon the acceptance of this manuscript.

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
