# Peer review of "Genetic Diversity and Population Structure of Jubaea chilensis, an Endemic and Monotype Gender from Chile, Based on SNP Markers"

_plants, 2022, doi:10.3390/plants11151959_

Round 1

Reviewer 1 Report

The manuscript is an original article on the genetic structure of the Chilean palm Jubaea chilensis. In general, the paper is well written, all the analyses have been carried out in the proper way and conclusions are supported by results. The Materials and method section is the one that requires major modifications. For instance, I do not understand the necessity of paragraphs 4.1 and 4.2. First, paragraph 4.1 (the description of the study system), if needed, should be inserted in the introduction (which includes all state-of-art information) and not in this section. Paragraph 4.2 is not clear to me: on what data such STRUCTURE analyses have been carried out? This M&M section could simply start at paragraph 4.3 but providing more data on the sampling sites. Some information relative to procedure and references is missing.

I have provided a list of things to fix, hoping that my suggestions will be of help in the revision process.

SPECIFIC COMMENTS

Introduction

Line 51 = allows

Line 51 - = it is not clear to me the statement “once high genetic diversity is necessary to species cope environmental changes”; please reformulate the sentence and evaluate the use of “once”.

Line 55 = please use the more common expression “cascade effect”

Line 57 = please replace the last comma with “and”

Line 70 = please remove the “a” before ‘keystone resource species’

Line 71 = cascade effect

Line 76 = in Central Chile

Line 81 = please replace “allied to” with “combined with”

Line 88 = what do you mean with “drupe with endocarp”? Probably there is an adjective missing (woody? Fleshy?). Furthermore, the expression “diclino-monoecious unisexual flowers” should be changed throughout the text with “diclino-monoecious” and only referred to species/taxa (flowers cannot be monoecious and diclino already means that the flower is unisexual).

Line 90 = what do you mean for “reduced survival”?

 Line 100 = please use the in-text citation reference style suggested by the journal

Line 103 = as above, please use the numbered-style for content in parenthesis

Results

Line 123 = the total number of reads was…

Line 124 = 101 bp is the mean read length? Please specify.

Line 132 = both suggesting

Line 133 = statistically significant is not if Fis is high, but if the values have a p < 0.05. Did the author perform such statistical test?

Line 142 = the info “(with Spearman correlation and 719 permutations)” is for M&M section; you do not need to specify the reference to Nei here.

Line 147-149 = it is not clear why some values in square brackets are separated by a dot and some by a semicolon. Please explain.

Line 150-153 = this is a statement for the discussion

Line 157 = in Table 1, I guess that the He, Ho and Fis values after the square brackets refer to the mean and the ones inside to min and max, right? If so, please explain it in the note. Same for Table 3 at line 170.

Line 175 = no need for reference here

Line 182 = Figure 2. despite is sometime reported in literature, you should not plot barplots at different K values because there are not at convergence and, therefore, they are wrong. Unless you don’t have two equally (or very close) K values, you should plot and discuss only the right one (in your case K=3).

Discussion

Line 195 = monoecious. Furthermore, as stated above, unisexual is redundant, so please leave just diclino-monoecious plant.

Line 196 = could explain

Line 197 = please specify what is such population

Line 204 = please remove (2009)

Line 204-205 = please carefully check the punctuation

Line 229-230 = this information is redundant

Line 238 = you can remove “eventual”

Line 265 = “of Extinction” is “at risk of Extinction”?

Line 274 = you probably meant “with” instead of “which”

M&M

Line 319 = there is missing information of population groups sampled; I do not understand why this STRUCTURE info is here, before GBS section.

Line 320 = what do you mean for “identifying six clusters”? Is this a result? If so, this is not the right section to report such info

Line 341 = I suggest removing “of bands” since DNA should appear as a single band if integer or as smear if degraded. Possible discrete bands could be RNA contaminations.

Line 352 = filtered on what? Please provide details of filtering procedure. If referring to information at lines 357-359, please insert it here. PLINK reference is missing.

Line 365 = reference for dartR is missing

Line 366 = one reference is for the package adegenet and it is misplaced

Line 373 = PCoA; please also specify that it is based on Nei genetic distance.

Line

Author Response

Comments and Suggestions for Authors R1

The manuscript is an original article on the genetic structure of the Chilean palm Jubaea chilensis. In general, the paper is well written, all the analyses have been carried out in the proper way and conclusions are supported by results. The Materials and method section is the one that requires major modifications. For instance, I do not understand the necessity of paragraphs 4.1 and 4.2. First, paragraph 4.1 (the description of the study system), if needed, should be inserted in the introduction (which includes all state-of-art information) and not in this section. Paragraph 4.2 is not clear to me: on what data such STRUCTURE analyses have been carried out? This M&M section could simply start at paragraph 4.3 but providing more data on the sampling sites. Some information relative to procedure and references is missing.

- Point 4.1.

Answer: We understand the referees’ concern, but it seems more appropriate to present the detailed description of the subject of study in the material & methods section than disrupting the introduction text with the bionomic traits. (Page 8, Lines 287-324).

- Point 4.2.

Answer: We agree and information of population groups sampled are presented on Figure 3 and Table. 1. We also removed the STRUCTURE info before presenting the data. (Page 9, Lines 326-331).

SPECIFIC COMMENTS

Introduction

- Line 51 = allows

Answer: Fixed, thanks. (Page 2, Line 51).

- Line 51 - = it is not clear to me the statement “once high genetic diversity is necessary to species cope environmental changes”; please reformulate the sentence and evaluate the use of “once”.

Answer: We rewrote this sentence and now it states: “The knowledge of genetic patterns in natural populations, in a context of global change, is essential for a more effective management of species, because a high genetic diversity is necessary to species cope environmental changes; and genetic erosion, together with demographic aspects, can increase extinction risk of natural populations [2].” (Page 2, Lines 50-54).

- Line 55 = please use the more common expression “cascade effect”

Answer: Done, thanks. (Page 2, Lines 56, 72).

- Line 57 = please replace the last comma with “and”

Answer: Fixed, thanks. (Page 2, Line 58).

- Line 70 = please remove the “a” before ‘keystone resource species’

Answer: Fixed, thanks. (Page 2, Line 71).

- Line 71 = cascade effect

Answer: Fixed, thanks. (Page 2, Line 56, 72).

- Line 76 = in Central Chile

Answer: Fixed, thanks. (Page 2, Line 77-78).

- Line 81 = please replace “allied to” with “combined with”

Answer: Fixed, thanks. (Page 2, Lines 83-84).

- Line 88 = what do you mean with “drupe with endocarp”? Probably there is an adjective missing (woody? Fleshy?).

Answer: We rewrote this sentence and now it states: "fruit is a drupe with an orange fleshy pulped-pericarp and a hard-lignified endocarp". (Page 2, Lines 89-90).

Furthermore, the expression “diclino-monoecious unisexual flowers” should be changed throughout the text with “diclino-monoecious” and only referred to species/taxa (flowers cannot be monoecious and diclino already means that the flower is unisexual).

Answer: Fixed, thanks. (Page 2, Line 88-89; Page 8, Line 289).

- Line 90 = what do you mean for “reduced survival”?

Answer: We rewrote this sentence and now it states: "This palm species presents low frequency of long-distance seed dispersal events, reduced survival of seedlings and its habitat has high anthropogenic intervention [25-26]". (Page 2, Lines 90-92).

- Line 100 = please use the in-text citation reference style suggested by the journal

Answer: Fixed, thanks.

- Line 103 = as above, please use the numbered-style for content in parenthesis

Answer: Fixed, thanks.

Results

- Line 123 = the total number of reads was…

Answer: Corrected, thanks

- Line 124 = 101 bp is the mean read length? Please specify.

Answer: No, it is the total read length sequenced by the Illumina HiSeq platform. We included this information (Page 3, Lines 126-127)

- Line 132 = both suggesting

Answer: Fixed, thanks.

- Line 133 = statistically significant is not if Fis is high, but if the values have a p < 0.05. Did the author perform such statistical test?

Answer: Thank you for pointing it out, we performed a statistical test. It is important to highlight that there is a close relationship between confidence intervals and significance tests. Specifically, if a statistic is significantly different from 0 at the 0.05 level, then the 95% confidence interval will not contain 0. We re-wrote this sentence as: “Inbreeding coefficient values (FIS) for all populations were statistically significant (overall value of 0.424, with 95% confidence interval not including zero), varying from 0.177 in Viña del Mar/ Valparaíso to 0.586 in Culimo, which also indicates high levels of inbreeding with little or no random mating (Table 1).” (Page 3, Lines 135-138).

- Line 142 = the info “(with Spearman correlation and 719 permutations)” is for M&M section; you do not need to specify the reference to Nei here.

Answer: Done, thanks. (Page 3, Lines 140-141).

- Line 147-149 = it is not clear why some values in square brackets are separated by a dot and some by a semicolon. Please explain.

Answer: Thank you for pointing it out. There is no difference between dot and semicolon, we corrected it.

- Line 150-153 = this is a statement for the discussion        

Answer: Fixed, thanks. (Page 3, Lines 149-153).

- Line 157 = in Table 1, I guess that the He, Ho and Fis values after the square brackets refer to the mean and the ones inside to min and max, right? If so, please explain it in the note. Same for Table 3 at line 170.

Answer: The values after the square brackets refer to the mean and the ones inside to the 95% Confidence Intervals. We add this information in the table captions. (Page 3, Lines 154-158).

- Line 175 = no need reference here for

Answer: Fixed, thanks.

- Line 182 = Figure 2. despite is sometime reported in literature, you should not plot barplots at different K values because there are not at convergence and, therefore, they are wrong. Unless you don’t have two equally (or very close) K values, you should plot and discuss only the right one (in your case K=3).

Answer: We changed it accordingly, thanks. (Page 6, Lines 180-185).

Discussion

- Line 195 = monoecious. Furthermore, as stated above, unisexual is redundant, so please leave just diclino-monoecious plant.

Answer: Fixed, thanks. (Page 2, Line 88-89; Page 8, Line 289).

- Line 196 = could explain

Answer: We rewrote this sentence and not it reads: Our results show that all studied populations of Jubae chilensis present low genetic diversity and high inbreeding. Given their biallelic heritage, the highest value of HE expected for SNPs is 0.5. However, the highest value observed for J. chilensis was less than a third of that value, indicating that these populations likely suffer from loss of genetic diversity. Null alleles are an issue for many marker types and could also result in a downward bias in estimated heterozygosity. Unfortunately, we are no able to estimate null alleles frequency and SNPs null alleles has not been describe in this species. Inbreeding coefficient was significant and high for all population, even though the species is diclino-monoescious, that favored crossing [44]. Interesting, the population with small number of specimens (CUL) had the lowest value of HE and the highest value of FIS. All these results are indicative that J. chilensis may be suffering from genetic erosion. (Page 6, Lines 193-203).

- Line 197 = please specify what is such population

Answer: The population is Culimo (CUL). We included this information: "The population with small number of specimens (CUL) had the lowest value of HE and the highest value of FIS. All these results are indicative that Jubae chilensis may be suffering from genetic erosion". (Page 6, Lines 200-202).

- Line 204 = please remove (2009)

Answer: Fixed, thanks.

- Line 204-205 = please carefully check the punctuation

Answer: Done, thanks

- Line 229-230 = this information is redundant    

Answer: It was removed.

- Line 238 = you can remove “eventual”

Answer: Fixed, thanks.

- Line 265 = “of Extinction” is “at risk of Extinction”?

Answer: OK, changed. (Page 1, Line 40).

- Line 274 = you probably meant “with” instead of “which”

Answer: Fixed, thanks.

M&M

- Line 319 = there is missing information of population groups sampled; I do not understand why this STRUCTURE info is here, before GBS section.

Answer: Thanks for pointing it out. We agree and information of population groups sampled are presented on Figure 3 and Table. 1. We also removed the STRUCTURE info before presenting the data. (Page 9, Lines 326-331).

- Line 320 = what do you mean for “identifying six clusters”? Is this a result? If so, this is not the right section to report such info

Answer: We've deleted this mention and the structure results are presented on results section. (Page 9, Lines 326-331).

- Line 341 = I suggest removing “of bands” since DNA should appear as a single band if integer or as smear if degraded. Possible discrete bands could be RNA contaminations.

Answer: OK, we fixed it. (Page 10, Lines 348-349).

- Line 352 = filtered on what? Please provide details of filtering procedure. If referring to information at lines 357-359, please insert it here. PLINK reference is missing.

Answer: Filtering processes were done with R package r2vcftools, a wrapper for VCFTools and not with PLINK. We corrected it. (Page 10, Line 363).

- Line 365 = reference for dartR is missing

Answer: OK, thanks for noticing it and we added to text. (Page 10, Lines 375).

- Line 366 = one reference is for the package adegenet and it is misplaced

Answer: OK, thanks for noticing it and we added to text. (Page 10, Lines 378).

- Line 373 = PCoA; please also specify that it is based on Nei genetic distance.

Answer: Fixed, thanks. (Page 10, Line 383).

Reviewer 2 Report

The peer-reviewed manuscript describes the population diversity and structure of Jubaea chilensis (Chilean palm), an endemic species of endangered plant. The manuscript is well-edited and should be of interest. I suggest minor improvements, taking into account the following:
No mention of null alleles. Although these are handled by the STRUCTURE software used, their possible impact should be highlighted in the results and/or discussion chapter, if only because the authors highlight the high absence of heterozygotes, which may be caused by the presence of null alleles in addition to the inbreeding mentioned.
Have SNP null alleles been described in this species? I suggest a search in the literature.

Specific comments:
Many references are incorrect. Where more than two sources are cited, and none have been mentioned before, they should be listed with a hyphen rather than separated by a comma.
in line 64: [7-10];

in line 75: [18-20];

in line 78: [21-23];

in line 89: [27-30];

in line 201: [45-50];

in line 208: [48-50];

In the sentence in lines 107-11: reference 26 is double cited. I suggest to remove it from the first place. It should be: "In spite that pollen dispersal can cover greater distance than seeds [36-38], in this case, its movement is limited because of the abundant flowers produced by just one individual which allow that pollinators visit contiguous flowers of the same individual, and visits to the flowers of other individuals are very infrequent events [26]."

You should avoid to cite more than 3 references in one place (eg. in line 201). In this case, please go to details.

Author Response

Comments and Suggestions for Authors R2

The peer-reviewed manuscript describes the population diversity and structure of Jubaea chilensis (Chilean palm), an endemic species of endangered plant. The manuscript is well-edited and should be of interest. I suggest minor improvements, taking into account the following:
No mention of null alleles. Although these are handled by the STRUCTURE software used, their possible impact should be highlighted in the results and/or discussion chapter, if only because the authors highlight the high absence of heterozygotes, which may be caused by the presence of null alleles in addition to the inbreeding mentioned.

- Have SNP null alleles been described in this species? I suggest a search in the literature.

Answer: Unfortunately, there is no information of null alleles described for this species. We included this information in the manuscript and now it reads: "Null alleles are an issue for many marker types and could also result in a downward bias in estimated heterozygosity. Unfortunately, we are no able to estimate null alleles frequency and SNPs null alleles has not been describe in this species”. (Page 6, Lines 193-203).

Specific comments:                               

- Many references are incorrect. Where more than two sources are cited, and none have been mentioned before, they should be listed with a hyphen rather than separated by a comma.
Answer: We fixed all references, thanks for noticing.

- In the sentence in lines 107-11: reference 26 is double cited. I suggest to remove it from the first place. It should be: "In spite that pollen dispersal can cover greater distance than seeds [36-38], in this case, its movement is limited because of the abundant flowers produced by just one individual which allow that pollinators visit contiguous flowers of the same individual, and visits to the flowers of other individuals are very infrequent events [26]." (Page 3, Lines 109-112).

Answer: Fixed, thanks.

- You should avoid to cite more than 3 references in one place (eg. in line 201). In this case, please go to details.

Answer: OK, we agree and fixed it.

Reviewer 3 Report

Dear Editor,

Here are my comments to the submitted manuscript ID: Manuscript ID: plants-1798411, which has been submitted to Plants, (Genetic Diversity and Population Structure of Jubaea chilensis, an Endemic and Monotype Gender from Chile, Based on SNPs Markers, by Jara-Arancio et al.)

The authors investigated population genetic diversity of Jubaea chilensis to assess the magnitude of the loss of genetic diversity as a result of isolation and low population sizes, using 1038 SNP markers. The authors claim that low levels of genetic diversity in the species, due to the high levels of inbreeding in populations and low level of gene flow.

Main comments

The paper is very interesting, the authors hypotheses and the biologically relevant questions are addressed clearly. The introduction is good. But when I read the materials and the methods section, I got frustrated. Thus, it’s very important, the authors MUST re-write this section very clear as materials and the methods section and not as an introduction (see line 278 to 314).

General comments

The introduction is ok.

Results

Page 3, line 139, I do not understand this sentence!

Page3, line 151-154, the authors mix the results and the discussion again.

Discussion

Page 7, line 198, All these results are indicative that Jubae chilensis may be suffering from genetic erosion, or may be founder effect?

Page 7, line 202, For instance, [45], shows high levels of inbreeding in populations of five species of the genus Pinanga. Cibrián-Jaramillo et al.  (2009) [46], in studies with microsatellites, declare that inbreeding is a result of the anthropic effect. [47], in turn, using nine ISSR markers, showed high levels of inbreeding in Attalea vitrivir in low abundance of populations. I do not understand it, please I require English correction.

Material and methods

Page 8, Must be re-write it again, see line 278 to 314.

Page 9, line 316, For the selection of population groups and individuals of J. chilensis to be sampled, we considered three parameters: i. populations with more than 50 individuals. I can see most number of individuals per populations are less than 27, can you explain that clearly.

We sampled the following populations groups: which populations and which groups?

We ran STRUCTURE v 2.3.4 [79,80,81], briefly, explain!

Finally, Do the SNPs analyses confirmed by repeating the analyses of a certain randomly selected individuals using these markers. The replicated profiles usually compared, and markers with more than 5% errors were removed from the datasets.

The authors have to improve their paper in resubmitted version and major work is needed. I believe the manuscript contains information that would be of interest to the readers, but need to be improved.

Author Response

Comments and Suggestions for Authors R3

Here are my comments to the submitted manuscript ID: Manuscript ID: plants-1798411, which has been submitted to Plants, (Genetic Diversity and Population Structure of Jubaea chilensis, an Endemic and Monotype Gender from Chile, Based on SNPs Markers, by Jara-Arancio et al.)

 The authors investigated population genetic diversity of Jubaea chilensis to assess the magnitude of the loss of genetic diversity as a result of isolation and low population sizes, using 1038 SNP markers. The authors claim that low levels of genetic diversity in the species, due to the high levels of inbreeding in populations and low level of gene flow.

Main comments

The paper is very interesting, the authors hypotheses and the biologically relevant questions are addressed clearly. The introduction is good. But when I read the materials and the methods section, I got frustrated. Thus, it’s very important, the authors MUST re-write this section very clear as materials and the methods section and not as an introduction (see line 278 to 314).

Answer: We appreciate the reviewer’s comments and the text was modified.

General comments

The introduction is ok.

Results

- Page 3, line 139, I do not understand this sentence!

Answer: We understand you concern and we reduced the sentence to make it clearer. We rewrote this sentence and now it states: "All FST comparisons between populations showed values above 0.049 and below 0.119, which indicates moderate genetic structuring (Table 2). Nei's genetic distance values were lower than 0.003 in all populations, also indicating that the populations may be genetically similar (Table 2). In the PCO, the first two axes explained 61.4% of the variance (35.0% and 26.4% respectively)". (Page 3, Lines 140-142).

- Page3, line 151-154, the authors mix the results and the discussion again.

Answer: We agree and we removed the sentence that discuss the results. 

We rewrote this sentence and now it states: "Low migration rates were observed (m = 0.010 a 0.022) except for these populations (source ® receiver): Cocalán ® Candelaria (0.237 [0.181-0.293]), Cocalán ® Culimo (0.280 [0.238-0.322]), Cocalán ® Petorca (0.266 [0.215-0.317]) and Viña del mar-Valparaiso ® Ocoa (0.268 [0.222-0.314]) (Table 3). The log likelihood values were comparable between BA3-SNPs runs, as the Bayesian deviance (Table S2)". (Page 3, Lines 149-153).

Discussion

- Page 7, line 198, All these results are indicative that Jubae chilensis may be suffering from genetic erosion, or may be founder effect?

Answer: We modified the sentence to make it clearer: "Our results show that all studied populations of J. chilensis present low genetic diversity and high inbreeding. Given their biallelic heritage, the highest value of HE ex-pected for SNPs is 0.5. However, the highest value observed for J. chilensis was less than a third of that value, indicating that these populations likely suffer from loss of genetic diversity. Null alleles are an issue for many marker types and could also result in a downward bias in estimated heterozygosity. Unfortunately, we are no able to estimate null alleles frequency and SNPs null alleles has not been describe in this species. Inbreeding coefficient was significant and high for all population, even though the species is diclino-monoescious that favored crossing [44]. Interesting, the population with small number of specimens (CUL) had the lowest value of HE and the highest value of FIS. All these results are in-dicative that J. chilensis may be suffering from genetic erosion". (Page 6, Lines 193-203).

- Page 7, line 202, For instance, [45], shows high levels of inbreeding in populations of five species of the genus Pinanga. Cibrián-Jaramillo et al.  (2009) [46], in studies with microsatellites, declare that inbreeding is a result of the anthropic effect. [47], in turn, using nine ISSR markers, showed high levels of inbreeding in Attalea vitrivir in low abundance of populations. I do not understand it, please I require English correction.

Answer: We modified the sentence to make it clearer. (Page 6, Lines 207-210). Now it reads: “For instance, Shapcott et al. [45] found high levels of inbreeding in populations of five species of the genus Pinanga. Cibrián-Jaramillo et al. [46] found a high inbreeding in populations of Chamaedorea ernesti-augusti, and, declare that the high inbreeding is a result of the anthropic effect.  Santos et al. [47], using nine ISSR markers, showed high levels of inbreeding and low abundance levels of Attalea vitrivir individuals.” (Page 8, Lines 208-212).

Material and methods

- Page 8, Must be re-write it again, see line 278 to 314.

Answer: It was modified. (Page 8, Lines 287-324).

- Page 9, line 316, For the selection of population groups and individuals of J. chilensis to be sampled, we considered three parameters: i. populations with more than 50 individuals. I can see most number of individuals per populations are less than 27, can you explain that clearly.

Answer: It was modified: A maximum of 30 individuals per population was analyzed. (Page 9, Line 328).

- We sampled the following populations groups: which populations and which groups?

Answer: It was modified. (Page 9, Lines 326-331).

- We ran STRUCTURE v 2.3.4 [79,80,81], briefly, explain!

Answer: We removed the STRUCTURE info before presenting the data. (Page 9, Lines 326-331).

- Finally, Do the SNPs analyses confirmed by repeating the analyses of a certain randomly selected individuals using these markers. The replicated profiles usually compared, and markers with more than 5% errors were removed from the datasets.

Answer: The SNPs analyzed had high coverage (at least 10x) and we only used SNPs that were present in more than 50% of the samples, to avoid artifacts. We believe that this is enough to guarantee that the data is robust enough. Using less samples would not give us certainty of the results, as most of the packages used need a large sample number to produce reliable results.

Round 2

Reviewer 3 Report

Dear Editor,

Here are my second comments to the submitted manuscript ID: Manuscript ID: plants-1798411, which has been submitted to Plants, (Genetic Diversity and Population Structure of Jubaea chilensis, an Endemic and Monotype Gender from Chile, Based on SNPs Markers, by Jara-Arancio et al.)

The authors improved their paper in resubmitted version but some work is needed it. The manuscript contains information that would be of interest to the readers. But the materials and methods section, specially paragraph, 4.1. Study organism as presented is not enough and need to be re-write again, see my previous comments, I already said; (it’s very important, the authors MUST re-write this section very clear as materials and methods section and not as an introduction).  English language and style are fine/minor spell check required.

Author Response

Dear Ms Irene Xia
Please find attached our revised manuscript(R2):
Our revised manuscript is attached: "Genetic diversity and population structure of Jubaea chilensis, an endemic and monotypic genus of Chile, based on SNP markers". We have addressed all the questions raised by all three reviewers and hope that our manuscript is now suitable for publication in Plants. Changes are highlighted in light blue in the article and changed references in pink.
Thank you for your manuscript submission. Please kindly revise your manuscript according to the reviewer 3's comments, and add a conclusion section to the manuscript as suggested by the academic editor.  Thanks again for your cooperation.
Answer:We add conclusion

Comments and Suggestions for Authors
Dear Editor,
Here are my second comments to the submitted manuscript ID: Manuscript ID: plants-1798411, which has been submitted to Plants, (Genetic Diversity and Population Structure of Jubaea chilensis, an Endemic and Monotype Gender from Chile, Based on SNPs Markers, by Jara-Arancio et al.)
The authors improved their paper in resubmitted version but some work is needed it. The manuscript contains information that would be of interest to the readers. But the materials and methods section, specially paragraph, 4.1. Study organism as presented is not enough and need to be re-write again, see my previous comments, I already said; (it’s very important, the authors MUST re-write this section very clear as materials and methods section and not as an introduction).  English language and style are fine/minor spell check required.
Answer:We rewrite this paragraph
